# Identification of the *Francisella novicida FTN_0096* as a factor involved in intracellular replication and host response

**Dhandy Koesoemo Wardhana**[1,2☯], **Takashi Shimizu**[1☯], **Kenta Watanabe**[1], **Akihiko Uda**[3], **Masahisa Watarai**[1]*

1 Laboratory of Veterinary Public Health, Joint Graduate School of Veterinary Medicine, Yamaguchi University, Yamaguchi, Japan, 2 Division of Veterinary Public Health, Department of Veterinary Science, Faculty of Veterinary Medicine, Universitas Airlangga, Surabaya, Indonesia, 3 Department of Veterinary Science, National Institute of Infectious Diseases, Shinjuku, Tokyo, Japan

☯ These authors contributed equally to this work.
* watarai@yamaguchi-u.ac.jp

## Abstract

*Francisella tularensis* is the causative agent of the zoonotic disease tularemia. We investigated a pathogenic factor of *F. tularensis* subsp. *novicida (F. novicida).* Accordingly, we established a novel infection model using HeLa cells. *F. novicida* usually infects macrophage lineage cells and less frequently epithelial cells. We successfully infected HeLa cells expressing the Fc receptor (HeLa–FcγRII cells) using *F. novicida* supplemented with mouse serum containing *F. novicida* antibodies. A total of 2,232 transposon mutants of *F. novicida* were screened to determine the relatively fewer cytotoxic strains of the HeLa–FcγRII cells, and 13 strains were thus isolated. Sequencing analysis of transposon insertion sites identified 13 genes, including *FTN_0096*. We focused on *FTN_0096*. Although the *F. novicida* wild-type strain proliferated in HeLa–FcγRII and THP-1 cells, the number of intracellular *FTN_0096* mutant decreased. *FTN_0096* mutant cannot escape from phagolysosomes in the initial phases of infection. Moreover, *FTN_0096* mutant was detected in the mitochondria and Golgi complex. These findings indicate the importance of *FTN_0096* of *F. novicida* for intracellular replication in the cells.

## Introduction

*Francisella tularensis* is a gram-negative coccobacillus that infects animals and humans through contact with infected arthropods or animals, consumption of contaminated food or water, or inhalation of bacterial aerosols [1,2]. Taxonomy of *F. tularensis* consists of some sub-species, including *tularensis* (type A), *holarctica* (type B), *mediaasiatica*, and *novicida*. Cottontail rabbits (*Sylvilagus spp.*) and ticks are the primary reservoirs of type A [3]. *F. tularensis* is a highly pathogenic facultative intracellular bacteria that can cause tularemia—a zoonosis disease [4]. *F. tularensis* is a dangerous

**Data availability statement:** All relevant data are within the manuscript and its Supporting information files.

**Funding:** TS: JSPS KAKENHI Grant Number 22K07054 MW: JSPS KAKENHI Grant Number 21H02360 The funders had no role in study design, data collection and analysis, decision to publish, or preparation of the manuscript.

**Competing interests:** The authors have declared that no competing interests exist.

pathogen that can incur a high risk of morbidity and mortality. *F. tularensis* causes typhoidal and pneumonic systemic tularemia, which has a fatality rate of up to 60% [5]. *F. tularensis* has been classified as a Tier 1 select agent by the Centers for Disease Control and Prevention (CDC) due to its extremely low infectious dose, potential for broad dissemination, and documented history of use as a biological weapon [6].

*F. tularensis* subsp. *novicida* is similar to *F. tularensis* but is a less virulent subspecies, as it can cause severe illness in mice but does not, usually, have much impact on the immunocompetence of humans. *F. novicida* is characterized as an uncommon opportunistic human infecting agent, capable of inducing morbidity and fatality in case of severe pathologies in either immunocompromised or medically weakened patients. *F. novicida*, which is regarded as a more environmentally adapted species, also shares several similarities to more virulent subspecies, including the LVS and Schu S4, a human virulent type A strain. *F. novicida* has a similar intracellular replication cycle within macrophages, characterized by swift phagosome escape and vigorous cytosolic replication, making it a commonly employed mouse model for studying the pathogenic mechanisms of tularemia [7,8].

For *F. tularensis* to effectively develop an infection and cause disease, it needs to replicate within the host cells. The intracellular lifecycle of *F. tularensis* has been observed *in vitro* with fixed and live-cell microscopy. Phagosome escape is necessary for intracellular replication, after which the organism must adapt to and replicate inside the host cytoplasm. When *F. tularensis* is internalized by a host cell, it enters macrophages via the phagocytosis process and evades lysosomal fusion, then the bacterium degrades the phagosome within 30 min and proliferates in the cytosol [9–11]. Phagosome escape is a crucial phase in *F. tularensis* lifecycle, and mutations in the *Francisella* pathogenicity island (FPI), which is responsible for encoding an alternate secretion pathway, lead to failed phagosome escape [12]. A characteristic of *F. tularensis* pathogenicity is its ability to remain immunologically silent during infection in both types of hosts (i.e., mice and people) and at the cellular level [13]. *F. tularensis* virulence factors, intracellular lifestyle, and interaction with different cell organelles are poorly known [14].

*F. novicida* infections rarely involve them due to their significantly lower infection efficiency in epithelial cell compared to macrophage-like cells. Research involving epithelial cell lines is markedly limited when compared to that of macrophage cell lines [15]. The efficiency of *F. novicida* uptake by macrophages is strongly influenced by opsonization. Opsonized bacteria, whether by serum or antibodies, are internalized by macrophages at levels that are 10-fold greater than their unopsonized counterparts [16]. Macrophages derived from the bone marrow of mice lacking Fc-gamma receptors (FcγR) show a 90% decrease in their ability to take up immunoglobulin G (IgG)-opsonized bacteria. Consequently, FcγR-mediated uptake could be a promising strategy for facilitating *F. novicida* infection in epithelial cells. Based on a previous research, HeLa–FcγRII cells were used as a new model to study how *F. novicida* infects epithelial cells [15].

The *FTN_0096* protein product is a conserved hypothetical membrane protein. *FTN_0096* is a member of the DUF1275 superfamily of proteins [17]. Members of this

family are found across all three domains of life, making them highly widespread. DUF1275 belongs to a large family characterized by a consistent structure of six—and occasionally seven—transmembrane segments (TMSs), which aligns with what's typically seen in half-sized proteins of the Major Facilitator Superfamily (MFS). Some of these proteins are encoded by genes located near what appears to be a YtcJ-like metallo–amido–hydrolase, suggesting a possible role in transporting peptides or other amido compounds, or exporting their breakdown products. However, none of these proteins have been functionally studied in detail, and their transport mechanisms remain unclassified [18].

In this study, we generated a transposon mutant library of *F. novicida* and screened it by infecting the mutant to the epithelial cells, HeLa cells expressing the Fc-gamma receptor (HeLa–FcγRII). This cell has the potential to enhance the efficiency of cell-based infection assays, such as large-scale genetic screening, and to offer novel insights into *Francisella* infection in epithelial cells, which has been difficult to analyze in phagocytic cells [15]. We focused on *FTN_0096* and discovered that the *FTN_0096* gene is a pathogenic factor in *F. novicida*.

## Materials and methods

### Bacterial strains and culture conditions

*F. novicida* U112 was sourced from the Pathogenic Microorganism Genetic Resource Stock Center (Gifu University). The bacterium was cultured aerobically at 37°C using either a chemically defined medium (CDM) [19] or a brain–heart infusion broth (Becton, Dickinson and Company, Franklin Lakes, NJ) enriched with cysteine (BHIc) [20] which included 1.5% agar (Wako Laboratory Chemicals, Osaka, Japan).

### Cell culture

HeLa–FcγRII cells were maintained in Dulbecco's Modified Eagle Medium (DMEM; Sigma-Aldrich, St. Louis, MO, USA) enriched with 10% heat-inactivated fetal bovine serum (FBS; Thermo Fisher Scientific, MA, USA) and incubated at 37°C in a humidified atmosphere with 5% $CO_2$. THP-1 cells, a human monocytic cell line, were cultured in RPMI 1640 medium (Sigma-Aldrich) enriched 10% heat-inactivated FBS under the same temperature and $CO_2$ conditions.

### Construction of a transposon mutant library

The transposon mutant library was developed using the Ez-Tn5 transposon system (Epicentre, Madison, WI). To construct pMOD3-FtKm, the multiple cloning sites of pMOD3 were digested with *Hind*III and *EcoR*I, and the kanamycin-resistance cassette of pKEK1140 [21] was inserted into these sites to generate pMOD3-FtKm. The transposon region of pMOD3-FtKm was then amplified by PCR, purified, and mixed with transposase following the protocol. This mixture was used to transform *F. novicida* via cryotransformation. The resulting transformants were plated on BHIc agar supplemented with 50 µg/mL of kanamycin.

### Screening of transposon mutant library

Microscopic observations were used to screen a mutant library of *Francisella novicida* to identify mutants with less cytotoxicity toward HeLa-FcγRII cells. To validate the cytotoxicity, lactate dehydrogenase (LDH) release assays were performed. HeLa–FcγRII cells ($5 \times 10^4$ cells/well) were seeded into 48-well tissue culture plate for 24 h. *F. novicida* strains preincubated with mouse serum containing specific antibodies were used at a multiplicity of infection (MOI) of 1.0. The cultured plates were centrifuged at $300 \times g$ for 10 minutes at room temperature, followed by incubation at 37°C for a designated period. After incubation, the cells were washed thrice with DMEM medium, and extracellular bacteria were eliminated by treating the cultures with gentamicin (50 µg/mL) for 1 hour. To measure LDH release as a marker of cytotoxicity, cells were incubated in the DMEM medium at 37°C for the specified time. The release of LDH into the culture supernatant was quantified using the LDH Cytotoxicity Detection Kit (Takara Bio, Shiga, Japan).

## Sequence analysis of transposon mutants

The plasmid pMOD3 contains the R6Kγ origin of replication from *Escherichia coli*. Genomic DNA from *F. novicida* transposon mutants was extracted using the PureLink Genomic DNA Mini Kit (Thermo Fisher Scientific) and digested with a combination of restriction enzymes *Xho*I, *Bgl*II, *EcoR*I, *Sal*I, *Not*I, *BamH*I, *Pst*I, and *Sph*I. The resulting DNA fragments were blunt-ended using the DNA-Blunting Kit (Takara Bio) and subsequently ligated using Ligation High Ver. 2 (Toyobo). The ligation products were used to transform One-Shot PIR1 Chemically Competent *E. coli* (Thermo Fisher Scientific). Transformants were selected on kanamycin-containing plates, and plasmid DNA was purified from the resistant colonies. Sequence analysis was performed using a primer described in the Ez-Tn5 transposon system manual.

## GFP-expression of *F. novicida* and complimentary strains

The GFP-expressing plasmid pOM5-GFP was constructed in accordance with the procedures published elsewhere [22]. This plasmid was introduced into the wild-type and *FTN_0096* strains of *F. novicida* via electroporation. To create pOM5–0096 or its complementary, *FTN_0096* gene along with its native promoter region (300-bp upstream) was cloned into the pOM5 vector. pOM5–0096 was then used to transform into the *FTN_0096* mutant of *F. novicida* via electroporation.

## Intracellular growth assay

THP-1 cells ($4 \times 10^5$ cells/well) were seeded into 24-well tissue culture plate and treated with 200 nM phorbol 12-myristate 13-acetate (PMA) for 48 h to induce differentiation. *F. novicida* strains were added at an MOI of 1. HeLa–FcγRII cells ($1 \times 10^5$ cells/well) were incubated overnight in 24-well tissue culture plates. *F. novicida* strains were incubated with mouse serum containing *F. novicida* antibodies and infected at an MOI of 1 to the HeLa–FcγRII cells. The THP-1 cells were washed thrice with RPMI1640 medium, HeLa–FcγRII cells were washed thrice with DMEM medium, and extracellular bacteria were eliminated by treating the cultures with gentamicin at the concentration of 50 µg/mL for 1 h. The cells were incubated in a fresh medium at 37°C for the desired time points. The cells were washed with phosphate-buffered saline (PBS) and then lysed with 0.1% Triton X-100 in CDM to determine the intracellular growth. Colony-forming units (CFUs) were determined by plating serial dilutions of the lysates on BHIc agar plates.

## Fluorescence microscopy

THP-1 cells ($4 \times 10^5$ cells per well) were seeded onto 12-mm glass coverslips placed in 48-well tissue culture plates and treated with 100 nM of PMA for 48 h. HeLa–FcγRII cells ($1 \times 10^5$ cells/well) were plated onto 12-mm glass coverslips in 24-well tissue culture plates and incubated overnight. THP-1 cells were infected with GFP-expressing *F. novicida* strains and incubated for the specified times. Separately, GFP-expressing *F. novicida* strains were opsonized with mouse serum and used to infect HeLa–FcγRII cells at a multiplicity of infection (MOI) of 1.0, followed by incubation at the indicated time points. The cells were stained with LysoTracker Red DND-99, MitoTracker® Deep Red FM, and BODIPY TR (Thermo Fisher Scientific) to visualize lysosomes, mitochondria, and Golgi complex as per the source's instruction manual. To detect lysosomal-associated membrane protein 1 (LAMP-1), the cells were fixed using the PLP Solution Set (Wako Laboratory Chemicals) containing 5% sucrose at 37°C for 1 h, followed by permeabilized with cold methanol for 10 s. The cells were treated with an anti-LAMP-1 antibody (ab25245, 1:100, Abcam) and subsequently stained with FITC-conjugated anti-rat IgG (1:1000, Abcam). The percentage of colocalizing bacteria was calculated manually by counting the total colocalized and non-colocalized bacteria in at least 100 infected cells. Confocal images were acquired using a FluoView FV100 confocal laser scanning microscope (Olympus, Tokyo, Japan).

## Statistical analysis

Statistical significance between groups was analyzed with the Bonferroni/Dunnett or the Student's *t*-test, $P < 0.01$ was considered statistically significant.

## Results

### Identification of *F. novicida* genes with reduced cytotoxicity to HeLa–FcγRII cells

We previously developed new infection model of *F. novicida* using HeLa cells expressing mouse FcγRII (HeLa-FcγRII). HeLa-FcγRII cells enhanced *F. novicida* infection efficiency in epithelial cells and enabled more detailed study of *Francisella*'s intracellular behavior [15]. Using this new infection model, we screened the *F. novicida* transposon mutant library to identify the new virulence factor of *F. novicida*. Considering that intracellular invasion of bacteria was mediated by antibodies and FcγRII in this model, we expected to elucidate new intracellular growth factors that could not be detected via conventional analysis using macrophages. We demonstrated in a previous study that *F. novicida* infection was cytotoxic to HeLa–FcγRII and induced cell death. Therefore, we made microscopic observations to screen the mutant library of *F. novicida* that lacked cytotoxicity (Fig 1A and 1B). Among 2,232 transposon mutants examined, 13 were identified as being less cytotoxic. LDH assays were conducted to validate these findings. When compared to the wild-type strain, these mutants reduced the extent of LDH release from HeLa–FcγRII cells (Fig 1C). The transposon-insertion sites in the mutant strains were identified through sequencing to identify the genes. We focused on mutants with the code number E12-3, where the sequencing results encoded *FTN_0096* and the function of *FTN_0096* were analyzed.

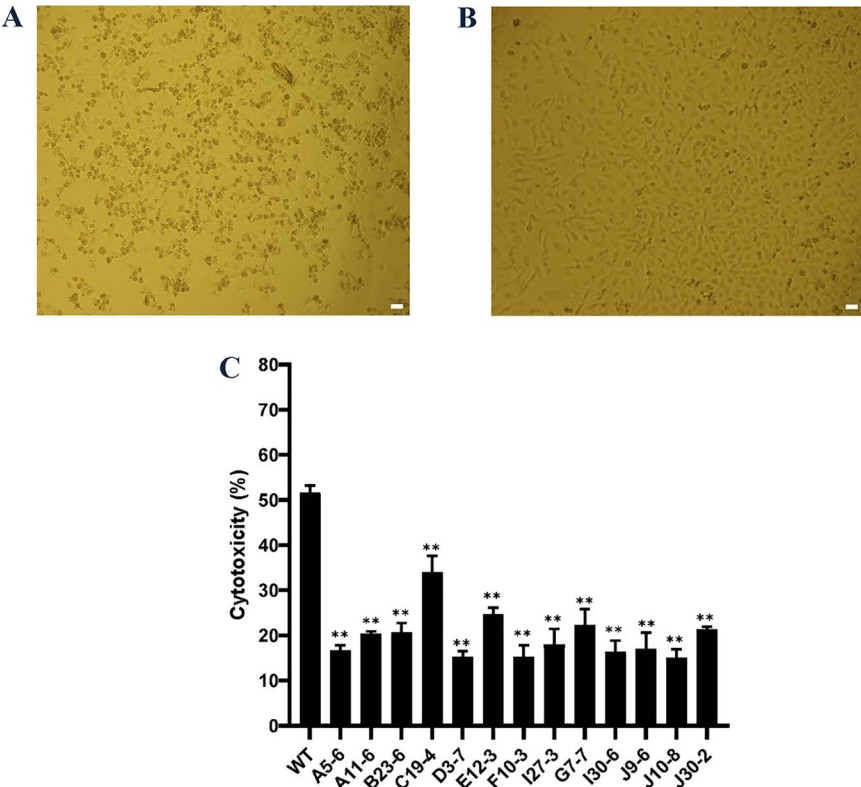

**Fig 1. Screening of the Transposon Mutant Library.** (A) HeLa–FcγRII cells were infected with *F. novicida* wild-type strain, preincubated with mouse serum containing *F. novicida* antibodies and incubated at 37°C for 24 h. Scale bar = 20 μm. (B) HeLa–FcγRII cells were infected with E12-3 transposon mutant of *F. novicida*, preincubated with mouse serum containing *F. novicida* antibodies, and incubated at 37°C for 24 h. Scale bar = 20 μm. (C) The transposon mutant library was screened using LDH cytotoxicity assay. HeLa–FcγRII cells were infected with the wild-type strain of *F. novicida* and transposon mutants of *F. novicida* preincubated with mouse serum containing *F. novicida* antibodies and incubated at 37°C for 24 h. LDH release was measured as an indicator of cytotoxicity. Data represent the mean and standard deviations of three identical experiments. Significant differences were evaluated in comparison to the wild-type strain using multiple comparison analyses as indicated by asterisks, **$P<0.01$.

### Effect of *FTN_0096* mutation on intracellular growth and cytotoxicity in HeLa–FcγRII cells

To analyze the function of *FTN_0096* on intracellular growth and cytotoxicity, we infected the *FTN_0096* transposon mutant of *F. novicida* to HeLa–FcγRII cells. The *FTN_0096* mutant (E12-3) induced a decreased LDH-release in comparison with the wild-type strain in HeLa–FcγRII cells (Fig 1C). To investigate the mechanism underlying cytotoxicity, we assessed the intracellular growth of the *FTN_0096* mutant in HeLa–FcγRII cells. While the wild-type *F. novicida* strain exhibited robust intracellular proliferation, the *FTN_0096* mutant showed significantly reduced intracellular growth in HeLa–FcγRII cells (Fig 2B). Replication was reinstated to the wild-type level in the complemented strains (Fig 2B). We also detected intracellular bacteria using GFP-expressing *F. novicida* strains. We examined the intracellular growth of the wild type and *FTN_0096* mutant at 12 and 24 h after infection in HeLa–FcγRII cells (Fig 2A). These results indicated that *FTN_0096* was essential for the cytotoxicity and intracellular growth of *F. novicida* using HeLa–FcγRII cells.

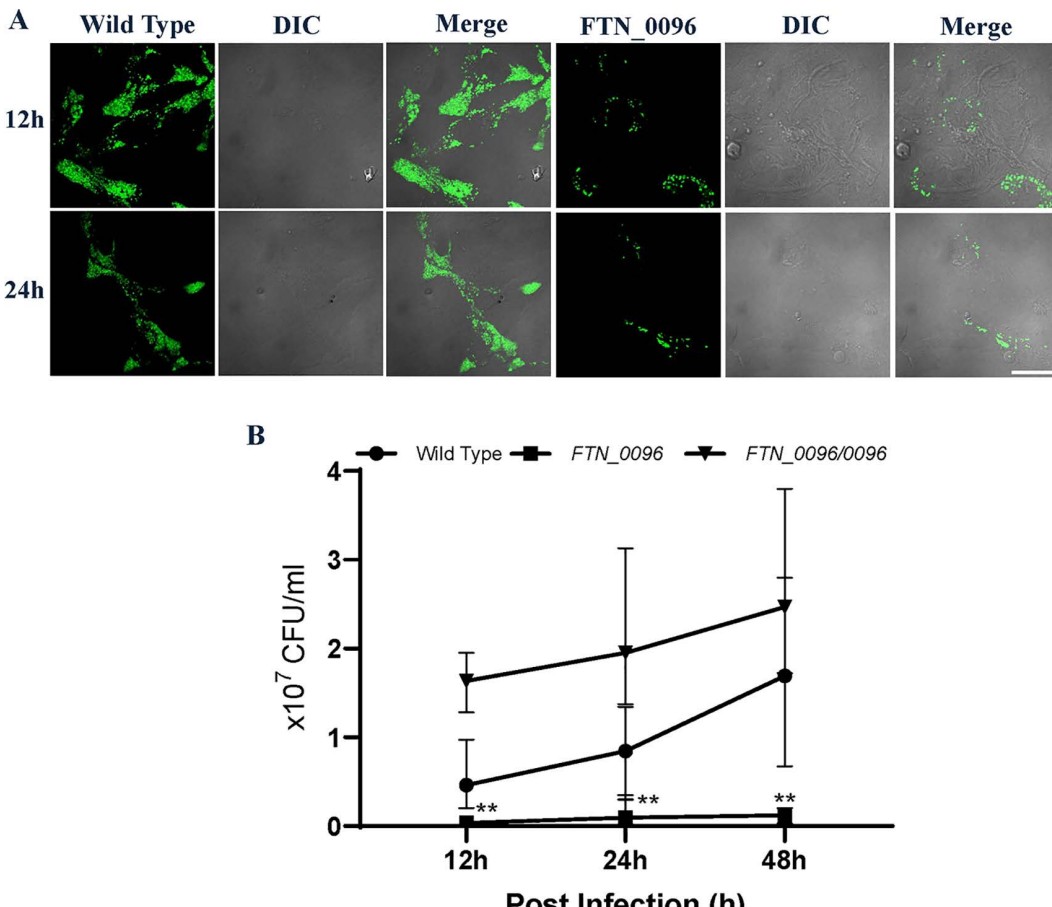

**Fig 2. Intracellular growth and cytotoxicity of the wild-type strain and *FTN_0096* mutant in HeLa–FcγRII cells.** (A) HeLa–FcγRII cells were infected with *F. novicida* wild-type strain and *FTN_0096* mutant, preincubated with mouse serum containing *F. novicida* antibodies at MOI = 1, and following treatment with gentamicin (50 μg/mL) for 1 h. The cells were fixed and examined at 12 and 24 h post-infection. Scale bar = 20 μm. (B) HeLa–FcγRII cells were infected with *F. novicida* wild-type, *FTN_0096* mutant, and complementary strain (*FTN_0096/0096*), preincubated with mouse serum containing *F. novicida* antibodies at MOI = 1, and following treatment with 50 μg/mL of gentamicin for 1 h. At 12, 24, and 48 h post-infection, cells were lysed with 0.1% Triton X-100, and plating serial dilutions on BHIc agar. Data represent the mean and standard deviations of three identical experiments. Significant differences were evaluated in comparison to the wild-type strain using multiple comparison analyses as indicated by asterisks, **$P < 0.01$.

## Escape of the bacterial cells from phagolysosomes in HeLa–FcγRII cells

To investigate the function of *FTN_0096* in intracellular growth, the ability of *F. novicida* strains to escape from phagolysosomes was measured. HeLa–FcγRII cells were infected with both the wild-type and *FTN_0096* mutant, and lysosomes were identified using an antibody against the lysosome marker LAMP-1 (Fig 3A). The wild-type strain exhibited continuous intracellular replication from 2 to 24 h post-infection, in line with the number of intracellular bacteria. In contrast, the number of *FTN_0096* mutant escalated until 12 h after the infection. However, the bacterial count diminished 24 h after the infection. In the case of infection with the wild-type strain, only a few numbers of bacteria showed colocalization with LAMP-1 in 27–42% of the cells (Fig 3B). On the other hand, the large numbers of *FTN_0096* mutant were observed with LAMP-1 in 70–79% cells at 2, 6, 12, and 24-h post-infection.

In order to further investigate the role of *FTN_0096* in the escape from phagolysosomes in HeLa–FcγRII cells, we observed the acidic organelles using Lysotracker. HeLa–FcγRII cells were infected with GFP-expressing *F. novicida* strains to examine their ability to escape from the phagolysosome during the early stages of infection (2 and 6 h post-infection). The cells were stained with LysoTracker and analyzed by confocal microscopy to visualize lysosomal localization. The co-localization of the Lysotracker of the wild type was observed in only 4–5% of infected cells, but the colocalization of *FTN_0096* mutant and Lysotracker was observed in 49–50% of the cells (Fig 3C). Few wild-type bacteria colocalized with lysosome, and several from the *FTN_0096* mutant colocalized with lysosome in HeLa–FcγRII cells (Fig 3D). These observations indicated that the wild-type bacteria evaded the phagolysosomes, whereas *FTN_0096* mutant failed to escape from them in HeLa–FcγRII cells.

## Effect of *FTN_0096* mutant on the mitochondria of HeLa–FcγRII cells

To further investigate the functions of *FTN_0096* in the mitochondria, HeLa–FcγRII were infected with the wild-type strains and the *FTN_0096* mutant. The mitochondria were visualized by using MitoTracker® Deep Red FM (Fig 4A) to observe whether the mitochondria can colocalize wild-type and *FTN_0096* mutant. The wild-type strain infected the HeLa–FcγRII cells, after which the intracellular bacterial cells were examined at 2–6 h post-infection. Nonetheless, a few cells colocalized with the mitochondria (Fig 4B). However, *FTN_0096* mutant was observed from 2 and 6 h after the infection, and several of the bacteria colocalized with the mitochondria, at approximately 75%–81% in HeLa–FcγRII cells (Fig 4B). These results suggest that *FTN_0096* mutant cannot modulate the mitochondria, rather *F. novicida* wild-type can modulate the mitochondria function and maintain their replication.

## Effect of *FTN_0096* mutant infection on the Golgi complex of HeLa–FcγRII cells

To analyze the function of *FTN_0096* on the Golgi complex, we infected HeLa–FcγRII cells with the wild-type strains and *FTN_0096* mutant. The Golgi complex was stained using BODIPY TR (Fig 4D) to observe whether the Golgi complex can colocalize with the wild-type strain and *FTN_0096* mutant. The wild-type strain infected the HeLa–FcγRII cells, and the intracellular bacterial cells were then examined at 2, 6, and 12 h after infection. Few bacteria colocalized with the Golgi complex (Fig 4C). Nevertheless, the *FTN_0096* mutant colocalized with the Golgi complex at all time points after the infection. Approximately 69%–84% of the bacteria from *FTN_0096* mutant colocalized with the Golgi complex in HeLa–FcγRII cells (Fig 4C).

## Intracellular growth of *FTN_0096* mutant in THP-1 cells

To explore the function of *FTN_0096* in THP-1 cells, we measured the intracellular growth of the *FTN_0096* mutant in THP-1 cells. While the wild-type strain exhibited robust intracellular proliferation, in contrast to the *FTN_0096* mutant, for which the numbers of intracellular growths reduced in THP-1 cells (Fig 5B). Replication was reinstated up to the wild-type level in the complemented strains (Fig 5B). We also detected intracellular bacteria using GFP-expressing *F. novicida*

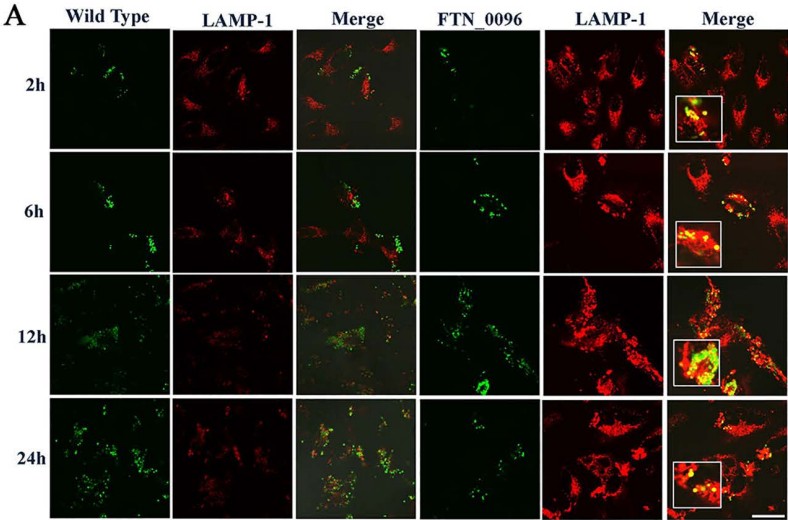

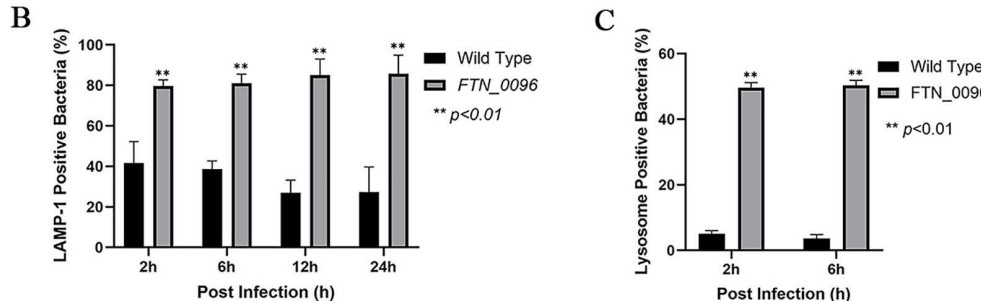

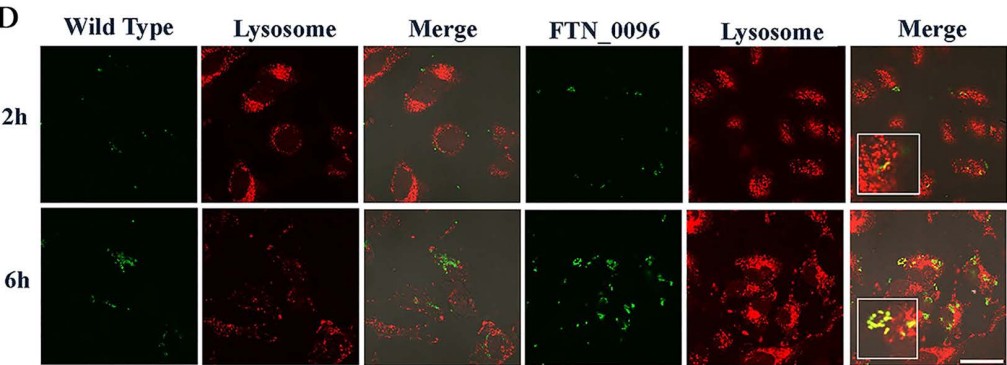

**Fig 3. Recognition of _F. novicida_ strains by lysophagosomes and lysosomes in HeLa–FcγRII cells.** (A) HeLa–FcγRII cells were infected with _F. novicida_ wild-type strain and _FTN_0096_ mutant preincubated with mouse serum containing _F. novicida_ antibodies at MOI = 1, and following treatment with gentamicin (50 μg/mL) for 1 h. At 2 to 24 h post-infection, cells were treated with an anti-LAMP-1 antibody and visualized using a TRITC-conjugated anti-rat IgG. Scale bar = 20 μm. (B) The percentage of _F. novicida_ wild-type strain and _FTN_0096_ mutant colocalized with LAMP-1. The data represent the mean and standard deviation from three identical experiments. Significant differences were evaluated in comparison to the wild-type strain using multiple comparison analyses as indicated by asterisks, **_P_<0.01. (C) The percentage of _F. novicida_ wild-type strain and _FTN_0096_ mutant colocalized with lysosomes. The data represent the mean and standard deviation from three identical experiments. Significant differences were evaluated in comparison to the wild-type strain using multiple comparison analyses as indicated by asterisks, **_P_<0.01. (D) HeLa–FcγRII cells were infected with _F. novicida_ wild-type strain and _FTN_0096_ mutant preincubated with mouse serum containing _F. novicida_ antibodies at MOI = 1, and following treatment with gentamicin (50 μg/mL) for 1 h. At 2 to 6 h post-infection, cells were stained with LysoTracker™ Red DND-99 to visualize lysosomes. Scale bar = 20 μm.

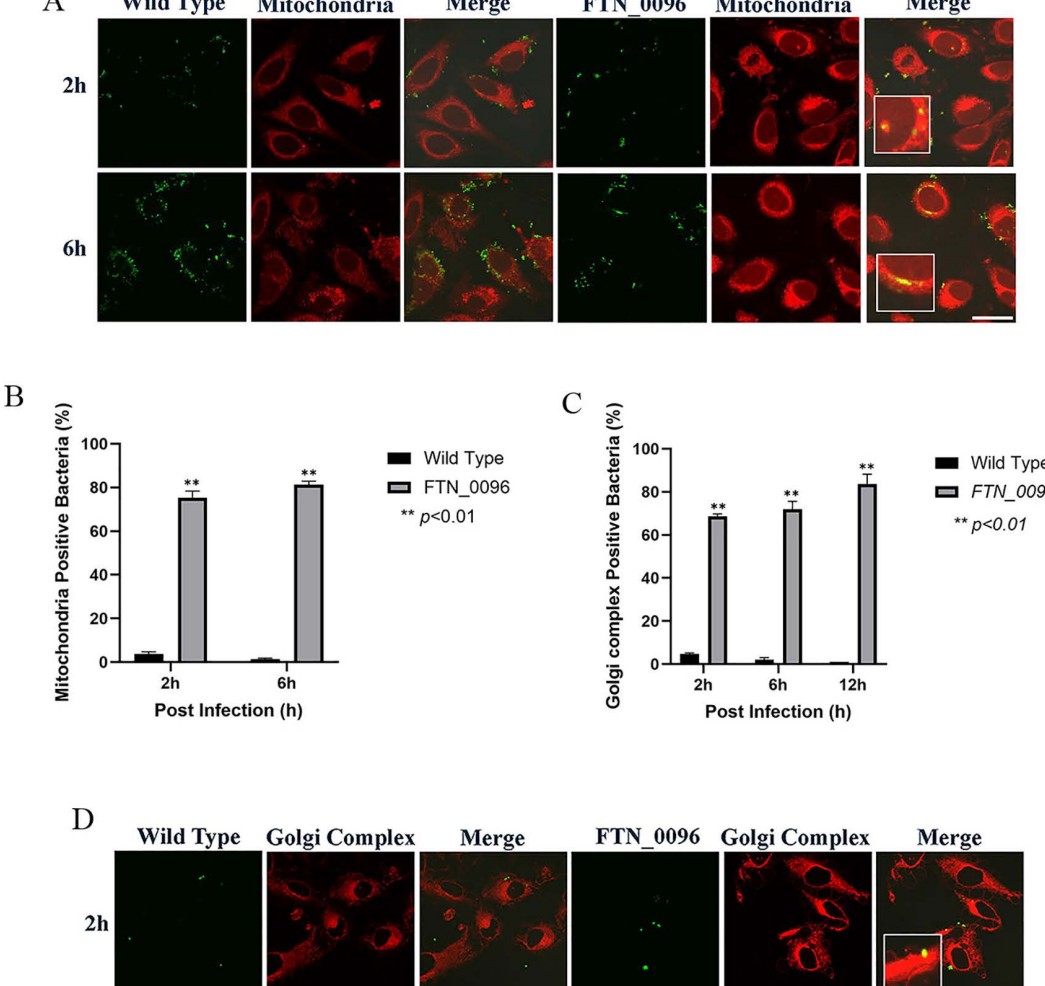

**Fig 4. Recognition of *F. novicida* strains by the mitochondria and Golgi complex in HeLa–FcγRII cells.** (A) HeLa–FcγRII cells were infected with *F. novicida* wild-type strain and *FTN_0096* mutant preincubated with mouse serum containing *F. novicida* antibodies at MOI = 1, and following treatment with gentamicin (50 μg/mL) for 1 h. Cells were stained with Mitotracker® Deep Red FM at 2 and 6 h post-infection. Scale bar = 20 μm. (B) Percentage of *F. novicida* wild-type strain and *FTN_0096* mutant that colocalized with the mitochondria. The data represent the mean and standard deviation from three identical experiments. Significant differences were evaluated in comparison to the wild-type strain using multiple comparison analyses as indicated by asterisks,**P<0.01. (C) The percentage of *F. novicida* wild-type strain and *FTN_0096* mutant that colocalized with the Golgi complex. The data represent the mean and standard deviation from three identical experiments. Significant differences were evaluated in comparison to the wild-type strain using multiple comparison analyses as indicated by asterisks, **P<0.01. (D) HeLa–FcγRII cells were infected with *F. novicida* wild-type strain and *FTN_0096* mutant preincubated with mouse serum containing *F. novicida* antibodies at MOI = 1, and treated with 50 μg/mL gentamicin. At 2, 6, and 12 h post-infection, cells were stained with BODIPY TR. Scale bar = 20 μm.

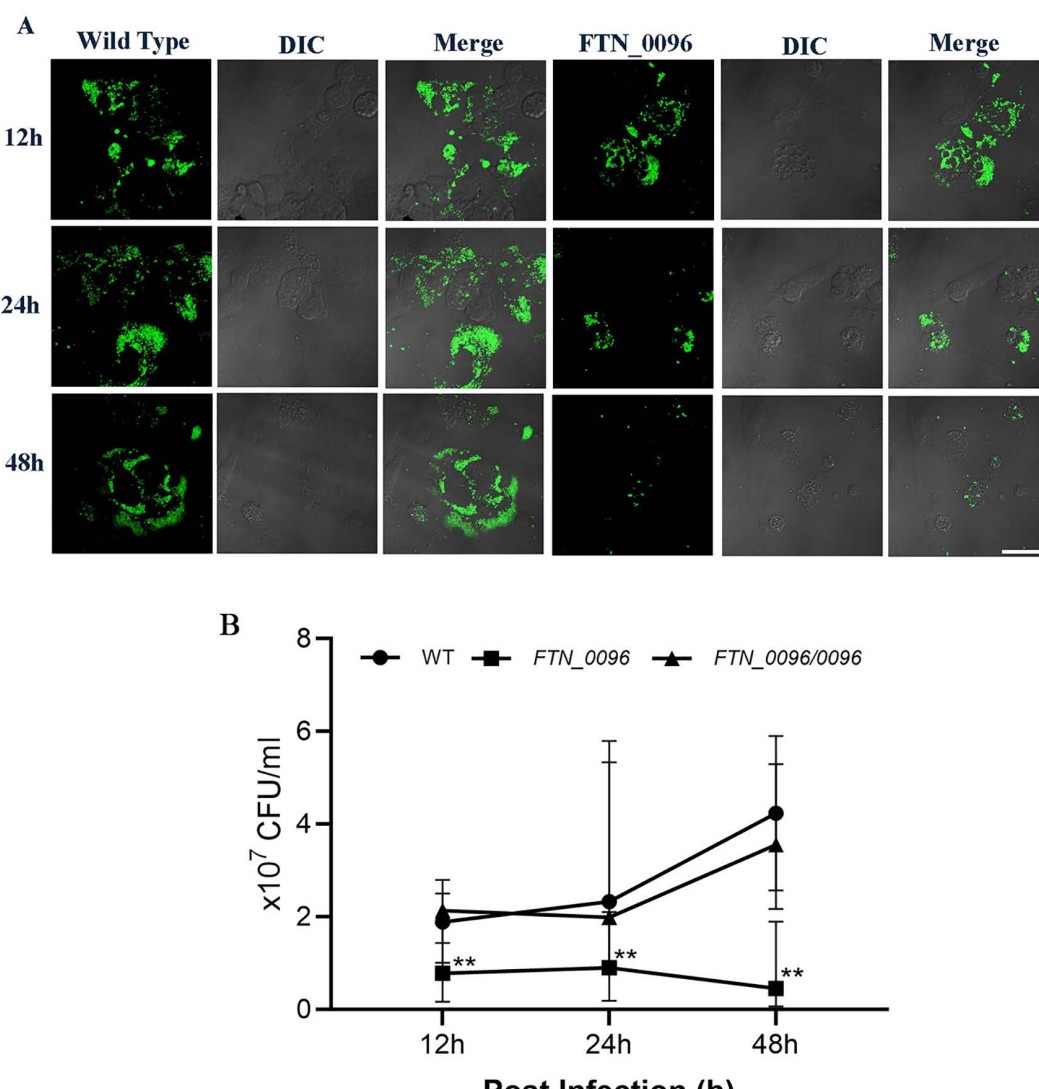

**Fig 5. Intracellular growth of the wild-type strain and *FTN_0096* mutant in THP-1 cells.** (A) THP-1 cells were infected with *F. novicida* wild-type strain and *FTN_0096* mutant at MOI = 1, following treatment with gentamicin (50 μg/mL) for 1 h. Cells were fixed and examined at 12, 24, and 48 h post-infection. Scale bar = 20 μm. (B) THP-1 cells were infected with *F. novicida* wild-type, *FTN_0096* mutant, and *FTN_0096/0096* at MOI = 1, and following treatment with gentamicin (50 μg/mL) for 1 h. At 12, 24, and 48 h post-infection, cells were lysed with 0.1% Triton X-100 and plating serial dilutions on BHIc agar. The data represent the mean and standard deviation from three identical experiments. Significant differences were evaluated in comparison to the wild-type strain using multiple comparison analyses as indicated by asterisks, **$P < 0.01$.

strains. Next, we observed the intracellular growth of the wild-type strain and *FTN_0096* mutant at 12, 24, and 48 h post-infection in THP-1 cells, respectively (Fig 5A). These results indicate the importance of *FTN_0096* for the intracellular growth of *F. novicida* in THP-1 cells.

## Escape of the bacteria from phagolysosomes in THP-1 cells

To assess ability of *F. novicida* to escape from the phagolysosome in THP-1 cells, we infected the wild-type and *FTN_0096* mutant into THP-1 cells and detection of lysosomes was performed using a LAMP-1–specific antibody

(Fig 6A). The wild-type strain showed robust intracellular proliferation from 2 to 24 h post-infection, in line with the number of intracellular bacteria. *FTN_0096* mutant proliferated until 12 h after infection; however, the bacterial count diminished 24 h post-infection. In the case of wild-type infection, only a few numbers of bacteria showed colocalization with LAMP-1 in 27–33% of infected cells (Fig 6B). However, a large number of colocalization of *FTN_0096* mutant with LAMP-1was observed in 70−79% of the cells at 2, 6, 12, and 24 h post-infection, indicating significant colocalization of the mutant with LAMP-1 (Fig 6B).

To further investigate the role of *FTN_0096* in the escape from phagolysosomes in THP-1 cells, the cells were infected with GFP-expressing wild-type strain and *FTN_0096* mutant and acidic organelles were stained using Lysotracker. The wild-type strain showed colocalization with Lysotracker only in 5−6% of the cells, whereas the *FTN_0096* mutant was colocalized with Lysotracker in 50−57% of the cells (Fig 6C). A few wild-type bacteria colocalized with lysotracker and several the *FTN_0096* mutant had colocalized with lysotracker in THP-1 cells (Fig 6D). These observations indicated that the wild-type bacteria evaded the phagolysosomes, whereas *FTN_0096* mutant failed to escape from them in THP-1 cells.

### Effect of *FTN_0096* mutant on the mitochondria of THP-1 cells

To further investigate the functions of *FTN_0096* in the mitochondria of THP-1 cells, THP-1 cells were infected with the wild-type strains and *FTN_0096* mutant. The mitochondria were visualized using MitoTracker® Deep Red FM (Fig 7A) to observe whether the mitochondria colocalized the wild-type strain and *FTN_0096* mutant. The wild-type strain infected the THP-1 cells, and intracellular bacterial cells were examined 2–6 h post-infection. Nonetheless, few cells colocalized with the mitochondria (Fig 7B). However, the *FTN_0096* mutant was observed from 2 to 6 h post-infection, and several cells (57%–77%) had colocalized with the mitochondria in THP-1 cells (Fig 7B).

### Effect of *FTN_0096* mutant infection on the Golgi complex of THP-1 cells

To analyze the functions of *FTN_0096* in the Golgi complex, we infected THP-1 cells with wild-type strains and *FTN_0096* mutant. The Golgi complex was stained with the BODIPY TR (Fig 7D) to observe whether the Golgi complex can colocalize with the wild-type strain and *FTN_0096* mutant. The wild-type strain infected the THP-1 cells, and intracellular bacterial cells were examined at 2, 6, and 12 h post-infection. Few bacterial cells colocalized with the Golgi complex (Fig 7C). Nevertheless, the *FTN_0096* mutant colocalized with the Golgi complex from 2, 6, and 12 h post-infection. Around 63%–82% of the bacterial cells from *FTN_0096* mutant colocalized with the Golgi complex in THP-1 cells (Fig 7C).

### Discussion

Intracellular bacteria, such as *Francisella*, possess mechanisms that help evade the host immune system and aid survival in the host cells. Macrophage-based cell lines, such as bone marrow-derived macrophage cells, phorbol myristate acetate (PMA)-differentiated U937, THP-1, and J774 cells have been widely used to study *Francisella* infection [7,23–25]. We previously developed an infection model of *F. novicida* using HeLa cells expressing mouse FcγRII (HeLa–FcγRII). When using the HeLa–FcγRII cells with antibody-treated bacteria, the efficiency of infection will be 100-fold greater compared to the normal HeLa cells [15]. This model promoted the discovery of novel factors vital for intracellular proliferation that could not be found through conventional analysis using macrophages. Accordingly, we constructed a transposon mutant library of *F. novicida* and screened it to identify mutants that were relatively less cytotoxic to the HeLa–FcγRII. As such, 13 mutants that showed decreased LDH release in HeLa–FcγRII cells were compared to the wild-type strain, and transposon-inserted genes were determined. Contrary to our initial expectations of finding novel virulence factors, the identified gene set largely consisted of previously characterized virulence genes such as *Slt*, *IglC*, *pdpA*, and *pdpB*. Although previous comprehensive analyses had implicated *FTN_0096* as a potentially important factor for intracellular proliferation within macrophages [17], its detailed function remains to be elucidated. Consequently, we selected *FTN_0096* for further investigation. In this research, we had difficulty making homologous recombination, which is one of the steps

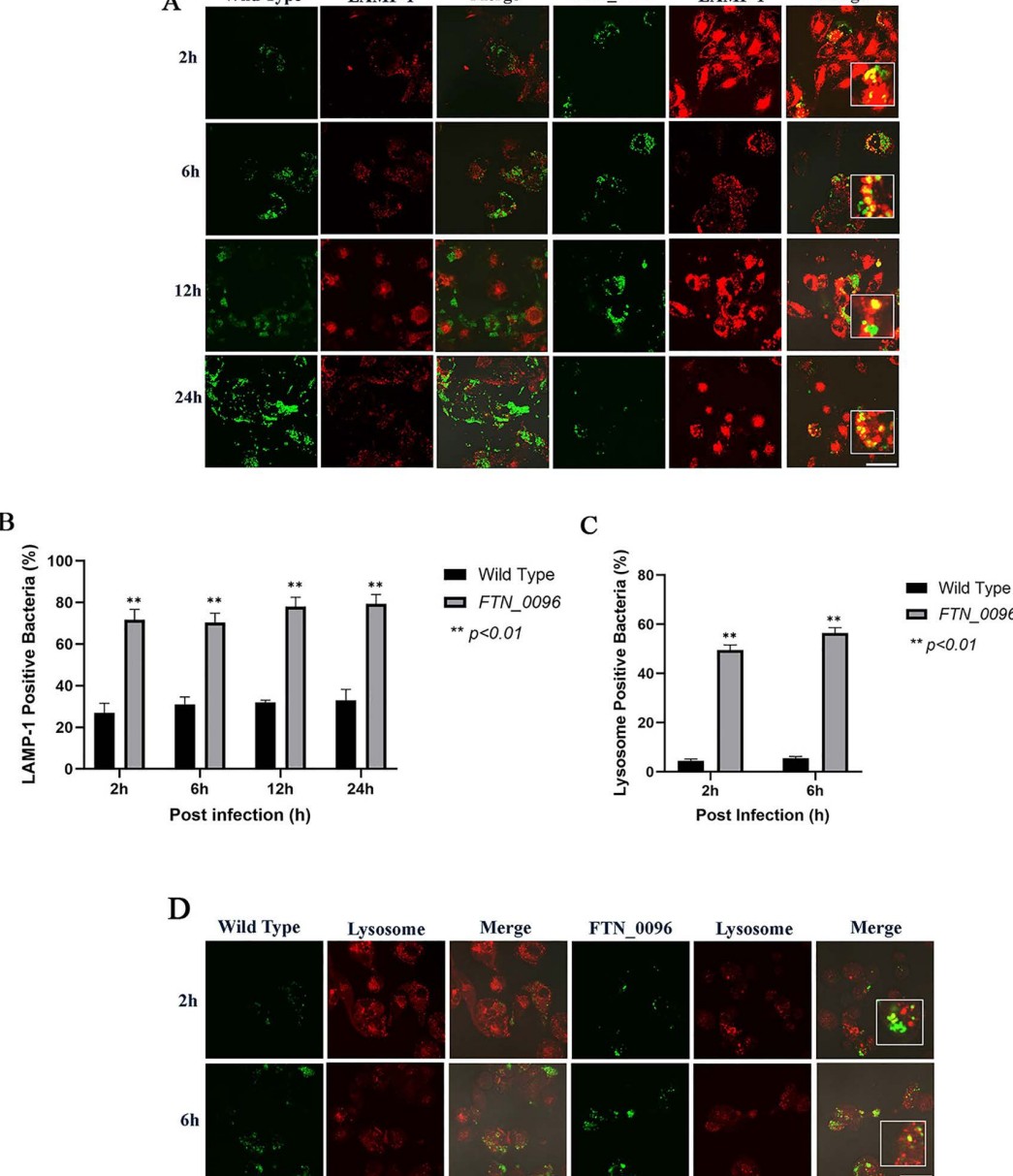

**Fig 6. Recognition of *F. novicida* strains by lysophagosomes and lysosomes in THP-1 cells.** (A) THP-1 cells were infected with *F. novicida* wild-type strain and *FTN_0096* mutant at MOI = 1, following treatment with gentamicin (50 μg/mL) for 1 h. Cells were treated with an anti-LAMP-1 antibody and stained with a TRITC-conjugated anti-rat IgG at 2–24 h after the infection. Scale bar = 20 μm. (B) The percentage of *F. novicida* wild-type strain and *FTN_0096* mutant colocalized with LAMP-1. The data represent the mean and standard deviation from three identical experiments. Significant differences were evaluated in comparison to the wild-type strain using multiple comparison analyses as indicated by asterisks, **$P < 0.01$. (C) The percentage of *F. novicida* wild-type strain and *FTN_0096* mutant colocalized with lysosomes. The data represent the mean and standard deviation from three identical experiments. Significant differences were evaluated in comparison to the wild-type strain using multiple comparison analyses as indicated by asterisks, **$P < 0.01$. (D) THP-1 cells were infected with *F. novicida* wild-type strain and *FTN_0096* mutant MOI = 1, following treatment with gentamicin (50 μg/mL) for 1 h. Cells were treated with LysoTracker™ Red DND-99 from 2 to 6 h after the infection. Scale bar = 20 μm.

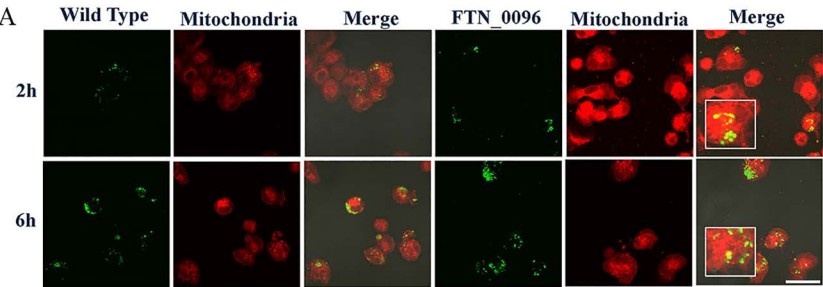

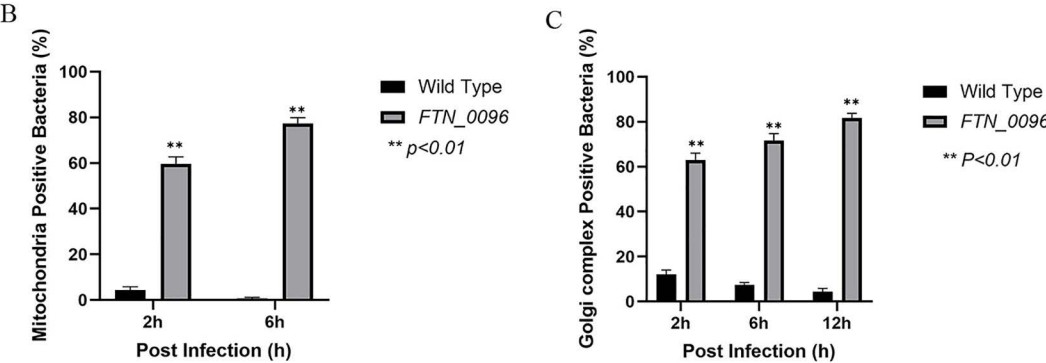

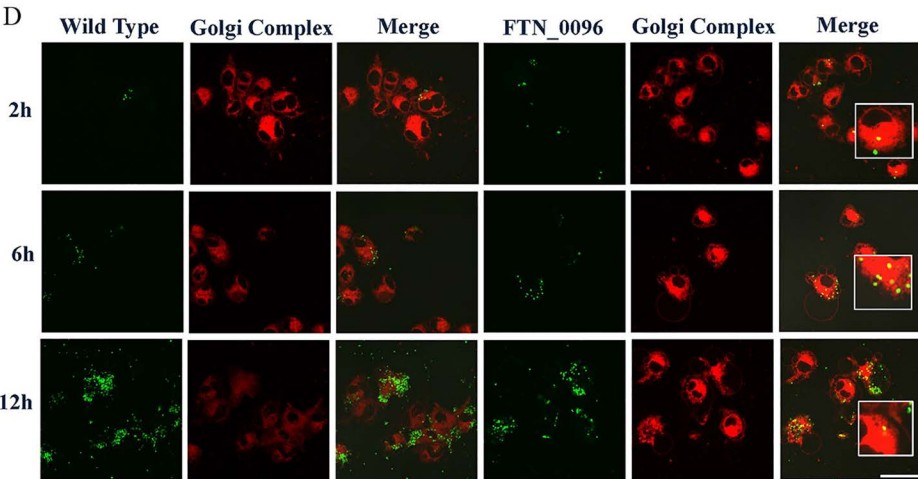

**Fig 7. Recognition of *F. novicida* strains by the mitochondria and Golgi complex in THP-1 cells.** (A) THP-1 cells were infected with *F. novicida* wild-type strain and *FTN_0096* mutant, MOI = 1, and following treatment with gentamicin (50 μg/mL) for 1 h. Cells were stained with the Mitotracker® Deep Red FM at 2 and 6 h post-infection. Scale bar = 20 μm. (B) The percentage of *F. novicida* wild-type strain and *FTN_0096* mutant that colocalized with mitochondria. The data represent the mean and standard deviation from three identical experiments. Significant differences were evaluated in comparison to the wild-type strain using multiple comparison analyses as indicated by asterisks, **$P < 0.01$. (C) The percentage of *F. novicida* wild-type strain and *FTN_0096* mutant that colocalized with the Golgi complex. The data represent the mean and standard deviation from three identical experiments. Significant differences were evaluated in comparison to the wild-type strain using multiple comparison analyses as indicated by asterisks, **$P < 0.01$. (D) THP-1 cells were infected with *F. novicida* wild-type strain and *FTN_0096*, MOI = 1, and following treatment with gentamicin (50 μg/mL) for 1 h. Cells were stained with BODIPY TR at 2, 6, and 12 h post-infection. Scale bar = 20 μm.

to make deletion mutants for this gene. It was suggested that complete gene deletion of *FTN_0096* might result in lethal. Therefore, we attempted to directly analyze *FTN_0096* from the transposon mutant library of *F. novicida*. A detailed examination of the dynamics of the *FTN_0096* transposon mutant was accordingly performed. The observed dynamics of the mutant were consistent in both HeLa–FcγRII cells and the macrophage-derived THP-1 cell line. These findings implied that the infection model using HeLa–FcγRII accurately reflected the behavior of macrophage-derived cells, making it a valuable platform for the study of virulence factors of *F. novicida*.

*FTN_0096* encodes a conserved hypothetical membrane protein and is a member of the DUF1275 superfamily, which includes proteins found in a wide range of bacterial species. While the function of DUF1275 remains unidentified, some of its members are encoded near genes for a predicted metallo–amido–hydrolase, suggesting a role related to the transport of peptides or other amido compounds, or in exporting of their hydrolysis products [18]. Nonetheless, the pathogenicity mechanism of *FTN_0096* of *F. novicida* remains unidentified.

*Francisella* enters the macrophages through phagocytosis and avoids lysosomal fusion by escaping from the resultant phagosome in 1–4 h [10]. Upon phagosomal escape, *Francisella* localizes to the host cytosol, an environment rich in nutrients and innate immune signaling components. *Francisella* manipulates the host cell metabolism to promote massive replication within the host cytosol [26]. The bacteria then re-enter the autophagosomes in the latter phase of infection [27].

*F. novicida* wild-type strain proliferated intracellularly well 12–48 h post-infection in HeLa–FcγRII and THP-1 cells. Nakamura et al. [28] demonstrated that the *F. novicida* wild-type strain proliferated intracellularly within THP-1 cells. In contrast, the *FTN_0096* mutant proliferated intracellularly until 12 h post-infection in HeLa–FcγRII and THP-1 cells but decreased 24 and 48 h post-infection. The rate of intracellular growth of the wild-type strain differed significantly from that in the *FTN_0096* mutant ($P<0.01$).

The *F. novicida* wild-type strain colocalized with acidic organelles (LysoTracker) 4%–5% and LAMP-1 at 27%–42% in HeLa–FcγRII. Similarly, in THP-1 cells, LysoTracker colocalization ranged from 5%–6%, while LAMP-1 levels were between 27%–33%. *F. novicida* wild-type strain can escape from phagosomes and replicate within the cytosol of HeLa–FcγRII and THP-1 cells. *FTN_0096* mutant was found within acidic compartments marked by LysoTracker during the initial phases of infection (2–6 h), and the mutant also localized with the lysosomal marker LAMP-1 during the later stages of the infection (12–48 h post-infection). The *FTN_0096* mutant colocalized with LysoTracker at 49%–50% and LAMP-1 at 80%–86% in HeLa–FcγRII cells, while in THP-1 cells colocalized by LysoTracker 50%–57% and LAMP-1 70%–79%. These results demonstrate that *FTN_0096* plays a critical role in the intracellular lifecycle of *F. novicida*, particularly in the ability to escape from the phagosome into the host cytosol. As successful escape into the host cytosol is essential for intracellular replication and the induction of cytotoxicity, the impaired ability of the *FTN_0096* mutant to escape from the phagosome likely accounts for its reduced intracellular proliferation and attenuated cytotoxic effects.

Lysosomes are membrane-bound organelles containing hydrolytic enzymes that facilitate the breakdown of diverse biological macromolecules. They exhibit considerable dynamic behavior and are capable of fusing with vesicles formed through endocytosis, auto-phagocytosis, and phagocytosis [29,30]. Lysosomes are essential for cellular waste disposal, digestion of foreign substances, recycling of cellular components, and the regulation of various signaling pathways. Dysfunction in lysosomal activity is associated with lysosomal storage disorders and can significantly impair cellular homeostasis [31].

Previous data have revealed that *F. novicida* can evade the phagosome and proliferate within the cytosol after the infection of mammalian cells. Intracellular pathogens have developed several methods to survive and evade the phagosome–lysosome process [32,33]. The role of host intracellular acidity in facilitating *F. novicida* reproduction remains unclear. However, past studies have demonstrated that some bacteria become sequestered within lysosomes where the alkalized pH of the intracellular environment can inhibit their replication [34–36]. Perhaps, *FTN_0096* may related to the mechanisms to survive in various pH of intracellular environment.

The *FTN_0096* mutant colocalized with mitochondria stained with MitoTracker® during the initial stages of infection (2–6 h), whereas the wild-type strains were not colocalized with the mitochondria. The *FTN_0096* mutant colocalized with the mitochondria (MitoTracker®) in HeLa–FcγRII cells at 75%–81% proportion and in THP-1 cells at 60%–77% proportion. These results indicate that the *FTN_0096* of *F. novicida* plays an important role in escape from the mitochondria and survive intracellularly. The mitochondrion is a vital organelle that facilitates several functions of the immune system, particularly via metabolic regulation, calcium homeostasis, and mitochondrial ROS (mtROS) production and degradation [37]. A distinct metabolic signature is mitochondrial clearance via mitophagy, which is important for activating the macrophage. Mitophagy, a form of autophagy specific to mitochondria, is designed to remove damaged mitochondria, either in response to stress or as a part of targeted mitochondrial clearing [38–41]. Previous results have shown the capability of *F. tularensis* SchuS4 to modulate mitochondrial processes early in infection by increasing mitochondrial activity and disrupting the metabolic transition from oxidative phosphorylation to glycolysis. Virulent *F. tularensis* manipulates host mitochondrial metabolism as a central strategy to restrict inflammation and regulate cell death for optimal bacterial replication [42]. *F. tularensis* also acts at different levels to maintain mitochondrial integrity and modulate cytosolic regulatory factors that alter the mitochondria to suppress caspase activity [43]. Caspases are essential components in apoptosis and are responsible for activating dismantling of the cell. Apoptosis is one of the host defense mechanisms that restricts the spread of pathogens. Apoptotic cells maintain their contents inside and are cleared by phagocytes. Some intracellular pathogens, such as *Rickettsia rickettsii*, *Legionella pneumophila*, *Salmonella enterica*, and *Chlamydia sp.*, prevent host cell apoptosis by acting on the mitochondria. Thus, the mitochondria are important targets for the regulation of apoptosis by intracellular pathogens [44,45]. Since *FTN_0096* mutant captured in mitochondria in this study, these findings suggested that *FTN_0096* may play an important role in intracellular growth by affecting the function of mitochondria.

To observe the activity of the *F. novicida* wild-type strain and *FTN_0096* mutant in the Golgi complex, the infected HeLa–FcγRII and THP-1 cells were stained using Bodipy TR. The result revealed that the colocalization of the Golgi complex to *F. novicida* wild-type strain was not observed, and *FTN_0096* mutant was observed. The percentage of wild-type strains that colocalized with the Golgi complex was 1%–5% in HeLa–FcγRII cells and 4%–12% in THP-1 cells. The *FTN_0096* mutant colocalized with the Golgi complex (Bodipy TR) in HeLa–FcγRII cells at 69%–84% and in THP-1 cells at 63%–82%. These results suggest that the *F. novicida* wild-type strain can escape from the intracellular events related to Golgi complex, but the *FTN_0096* mutant was detected within the Golgi complex, indicating that this mutant cannot escape from the events related to Golgi complex.

The Golgi complex, a vital cytoplasmic organelle, is the primary site for the post-translational modification, transport, and sorting of lipids and proteins synthesized by the endoplasmic reticulum. This complex ensures their transport to designated cellular locations via cytophagy and cytotoxicity. The typically compact structure of the Golgi complex can be disrupted, leading to various levels of disorganization and unstacking because of the cytotoxic agents. These changes in Golgi structure are often linked to problems with the transport and secretion of several essential hepatic glycoproteins. Furthermore, it engages in other biological processes, such as cell signaling, division, and apoptosis. Because of this primary function of the Golgi complex, pathogens try to develop strategies to disrupt and utilize these processes to facilitate the proliferation of pathogens [46–48]. The Golgi complex is essential for the induction of type I interferon (IFN-I) in response to several stimuli, such as *Francisella* infection, Toll-like receptors activation, and viral infection [49]. IFN-I is a class of cytokines that are deemed essential for viral and bacterial infections in regulating innate and adaptive immunity [50]. Previous research has suggested that IFN-I can impair the host's ability to clear intracellular bacteria, such as *Listeria monocytogenes* [51], *Mycobacterium tuberculosis* [52], and *F. tularensis* [53]. Intracellular bacteria, which are similar to viruses, proliferate within the cytoplasm and can trigger IFN-I products through intracellular pathways. Intracellular bacterial infections can create a cellular environment that mimics viral infection, potentially helping pathogens evade immune surveillance. IFN-I helps limit neutrophil swarming as a mechanism to mitigate excessive tissue injury, whereas intracellular bacteria may utilize this process to elude immunological detection by neutrophils. Disruption of neutrophil swarming,

caused by IFN-α, has been recognized as a significant contributor to increased susceptibility to bacterial infections [50]. These findings may suggest that *FTN_0096* may affect the function of Golgi complex to survive intracellularly.

Although the detailed function of *FTN_0096* in the mitochondria and the Golgi complex is not yet fully understood. The *FTN_0096* mutant showed increased colocalization with these organelles, indicating a failure to escape host cell defenses. *FTN_0096* may modulate host intracellular trafficking or stress response pathways that are regulated by these organelles, thereby promoting bacterial survival by interfering with host defense mechanisms. Comprehensive research is needed to determine whether *FTN_0096* directly modulates mitochondrial dynamics or Golgi-associated pathways during infection.

In conclusion, we screened a transposon mutant library using HeLa–FcγRII cells and identified *FTN_0096* as a pathogenic factor of *F. novicida*. *FTN_0096* is required for intracellular replication in the cells. *FTN_0096* contributes to bacterial survival by promoting escape from the phagosome and avoiding host organelle-mediated defenses, including those involving the mitochondria and Golgi complex. Further studies are however warranted to establish the detailed pathogenicity mechanisms of *F. novicida*, which may provide a basis for comprehending how *Francisella* expresses its pathogenicity. Moreover, the detailed mechanisms of *FTN_0096* can contribute to developing vaccines that can prevent diseases caused by *Francisella*.

## Supporting information

**S1 Data. Raw data for all figures and statistical analyses.** This file contains the raw measurements for the experiments described in the manuscript, with separate tabs corresponding to each figure.
(XLSX)

## Author contributions

**Conceptualization:** Takashi Shimizu.

**Data curation:** Dhandy Koesoemo Wardhana, Takashi Shimizu.

**Formal analysis:** Dhandy Koesoemo Wardhana, Takashi Shimizu.

**Funding acquisition:** Takashi Shimizu, Masahisa Watarai.

**Investigation:** Takashi Shimizu, Kenta Watanabe.

**Methodology:** Takashi Shimizu.

**Project administration:** Takashi Shimizu.

**Resources:** Takashi Shimizu, Kenta Watanabe, Akihiko Uda.

**Supervision:** Takashi Shimizu, Kenta Watanabe, Masahisa Watarai.

**Validation:** Takashi Shimizu.

**Visualization:** Dhandy Koesoemo Wardhana, Takashi Shimizu.

**Writing – original draft:** Dhandy Koesoemo Wardhana.

**Writing – review & editing:** Takashi Shimizu, Kenta Watanabe, Akihiko Uda, Masahisa Watarai.

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
