## [Decision Letter · Decision Letter 0]

Dear Dr %:LAST_NAME%,

Thank you for submitting your manuscript to PLOS ONE. After careful consideration, we feel that it has merit but does not fully meet PLOS ONE’s publication criteria as it currently stands. Therefore, we invite you to submit a revised version of the manuscript that addresses the points raised during the review process.

Your manuscript has been reviewed by three experts in your field and a minor revision is suggested. Please follow their comments and make all necessary revision or give a rebuttal.

We look forward to receiving your revised manuscript.

Kind regards,

Yung-Fu Chang

Academic Editor

PLOS ONE

Journal Requirements:

2. Thank you for stating the following financial disclosure: [TS: JSPS KAKENHI Grant Number 22K07054 MW: JSPS KAKENHI Grant Number 21H02360]. 

Reviewers' comments:

Reviewer's Responses to Questions

**Comments to the Author**

1. Is the manuscript technically sound, and do the data support the conclusions?

Reviewer #1: Yes

Reviewer #2: Partly

Reviewer #3: Yes

2. Has the statistical analysis been performed appropriately and rigorously?

Reviewer #1: Yes

Reviewer #2: Yes

Reviewer #3: Yes

3. Have the authors made all data underlying the findings in their manuscript fully available?

Reviewer #1: Yes

Reviewer #2: Yes

Reviewer #3: Yes

4. Is the manuscript presented in an intelligible fashion and written in standard English?

Reviewer #1: Yes

Reviewer #2: Yes

Reviewer #3: No

Reviewer #1: The authors investigated the importance of F. novicida FTN_0096 as an essential factor for intracellular replication of F. novicida within host cells. In this study, the authors showed that an F. novicida FTN_0096 mutant generated by transposon mutation could not escape from phagolysosomes in the initial phases of infection and was subsequently digested within the lysosomes after being captured by autophagosomes. All experimental methods were well designed, scientific results and the logical discussion were well described. This study may help to understand the virulence factors required for intracellular replication of F. novicida within phagocytes after infection, as well as and phagocytic pathways involved in F. novicida entry into host cells. Moreover, it may help to understand the basic aspects of immunological responses and the development of preventive tools in Francisella infection in humans and animals. Considering that Francisella infection is an important zoonosis by an intracellular parasitic pathogen, this study suggests that F. novicida FTN_0096 is an essential factor for intracellular replication and pathogenicity of F. novicida. However, this reviewer believes that the manuscript must be improved in several areas.

1. The abstract is too long and needs to be revise. The authors should delete the general introduction in abstract. In particular, lines 20 to 25 ‘Francisella tularensis is the causative agent of the zoonotic disease tularemia. F. tularensis is a gram-negative, facultative intracellular bacteria. F. tularensis subsp. (F. novicida) is a facultative intracellular pathogen that proliferates within macrophages; it has the minimum incidence of virulence in humans but is pathogenic in mice. Furthermore, F. novicida shares considerable similarities with highly virulent subspecies and is extensively utilized as a surrogate in the investigation of Francisella.’ should be deleted or moved to the Introduction section.

2. The format of whole manuscript must be checked. Especially, the titles of figures and references (Italic font, etc.) must be checked before submission.

3. The authors found that F. novicida FTN_0096 mutant could not escape from the phagosome and FTN_0096 is an essential factor for the intracellular replication of F. novicida within host cells. A more detail description of this finding is needed in the Discussion section to help readers better understand the significance of the study.

Reviewer #2: General Comments

In this paper, the authors investigated the virulence mechanism of F. novicida using a novel infection model, HeLa–FcγRII. They demonstrated that FTN_0096 is involved in the intracellular growth of F. novicida in both HeLa cells and THP-1 cells. While the reviewer agrees that FTN_0096 contributes to the intracellular survival of F. novicida, some parts of the manuscript should be revised for clarity and readability.

Specific Comments

Figs.: "FTN_0096" is probably used as an abbreviation for the FTN_0096 mutant (E12-3), but this is not described anywhere. Please revise this point.

Line 1: The involvement of FTN_0096 in F. novicida pathogenicity has already been reported. Moreover, it is unclear whether FTN_0096 itself is cytotoxic. Therefore, the reviewer suggests considering a revision of the title.

Lines 28, 77: The reviewer does not understand why the authors mentioned the difference in genetic manipulability between HeLa cells and macrophages. Since no gene editing was done in this study, this explanation may be unnecessary.

Lines 35-37, 260-266, 491: Why do the authors consider that the FTN_0096 mutant is captured by autophagosomes at the late stage of infection if this mutant fails to escape from phagolysosomes?

Lines 48-87, 194, 252, 429-555: In this context, what does “Francisella” refer to? It may be used to indicate all F. tularensis subspecies, rather than the genus Francisella. Please clarify the definition.

Lines 100, 196-199: Why were HeLa–FcγRII cells used for screening? In the Introduction, these cells are merely described as a new model. Please explain the advantages of this model and clarify why it was selected for the screening.

Line 118: Please describe the screening procedure.

Line 154: The use of "strains" may not be appropriate. Please correct.

Line 159: Please provide the full name of PMA.

Line 161: Please revise the sentence to indicate that mouse serum contains anti-F. novicida antibodies.

Lines 204, 440: Please revise "FcR HeLa cells" to " HeLa–FcγRII cells".

Lines 211–213: The reviewer infers that the cells were infected individually with each mutant, and that panel B is a representative image of the screening assay. If this is correct, please revise the sentence.

Lines 222–223: Using "mutation" instead of "mutant" in this title might be more appropriate.

Line 224: Maybe, a modification like "To analyze the function of FTN_0096" is appropriate.

Lines 252–253: The meaning of this sentence is unclear. You might revise the phrase "particularly the mechanism of FTN_0096 escapes" to "particularly in the escape" for clarity.

Lines 335-345: The authors describe the cytotoxicity of F. novicida against THP-1 cells; however, the results only demonstrate intracellular growth. Is F. novicida cytotoxic to THP-1 cells, and is FTN_0096 involved in this process?

Lines 358-359: This sentence is grammatically incomplete. Please revise it.

Which results are the values "50%-75%" and "59%-84%" derived from?

Lines 491-494: The reviewer could not follow the intention of these sentences. What do the phrases "These findings," "The same results," and "These results" refer to?

Lines 501-502: The reviewer could not understand the necessity of this sentence. Could you consider removing it?

Lines 518, 545: Please clarify what "These results" refer to.

Lines 528-529: Are there any contributions of cytophagy or cytotoxicity to the transport function of the Golgi complex? If so, please provide a citation.

Lines 549, 550: This sentence might not match the purpose and results of this manuscript. You might revise this sentence like "In conclusion, we screened a transposon mutant library using HeLa–FcγRII cells and identified FTN_0096 as a pathogenic factor of F. novicida ".

Lines 550-552: Are the authors suggesting that FTN_0096 mainly facilitates the escape of F. novicida from the capture by mitochondria and the Golgi complex? Is there any hypothesis about how FTN_0096 contributes to this escape, and how it is involved in the intracellular growth and cytotoxicity of this bacterium? A discussion of these points might help clarify the manuscript.

Reviewer #3: General Comments

In this paper, the authors identified a virulence factor of F. novicida using a novel infection model, HeLa–FcγRII. They demonstrated that FTN_0096 plays a critical role in the intracellular replication of F. novicida. While the reviewer concurs with this finding, certain sections of the manuscript require revision.

Specific Comments

Methods: The method for calculating the colocalization rate of bacteria and each cell organelle is not provided. Please revise this point.

Line 159: Please provide the full name of PMA.

Line 161: Please revise "with mouse serum" to "with mouse serum contains anti-F. novicida antibodies".

Lines 204, 440: Please replace "FcR HeLa cells" with "HeLa–FcγRII cells".

Lines 200-202, 211–213: In Fig. 1B, the bacterial strain that infected the HeLa-FcγRII cells is not clearly described. The reviewer assumes that HeLa-FcγRII cells were infected with the FTN_0096

mutant strain; is this correct? Please revise the legend of Fig. 1B to include more detailed information about the bacterial strain employed.

Lines 211, 213: "Scale bar = 20 μm" is missing a period, please correct.

Lines 335-345, 347-348: The authors describe the 'cytotoxicity of the FTN_0096 mutant in THP-1 cells'; however, the cytotoxicity assay was only performed using HeLa–FcγRII cells. Furthermore, results of the cytotoxicity assay for THP-1 cells are neither presented nor discussed in the manuscript. Therefore, the term 'cytotoxicity' should be removed from these sentences.

Lines 491-494: The reviewer was unable to clearly understand the intent of these sentences. It is unclear what is meant by the phrases 'These findings,' 'The same results,' and 'These results.'

Lines 549-550: The reviewer finds this sentence confusing. In the context of this study, it would be more appropriate to revise 'and identified FTN_0096 as a pathogenic factor of F. novicida' instead of 'to identify FTN_0096 as a pathogenic factor of F. novicida'.

**Do you want your identity to be public for this peer review?** For information about this choice, including consent withdrawal, please see our Privacy Policy

Reviewer #1: No

Reviewer #2: No

Reviewer #3: No

---

## [Author Response · Author response to Decision Letter 1]

7 Jul 2025

July 4, 2025

Dear Mr. Yung-Fu Chang

Academic Editor

Plos One

Thank you for giving us the opportunity to strengthen our manuscript with your valuable comments and queries. We have worked hard to incorporate your feedback and hope that these revisions persuade you to accept our submission.

• Academic Editor

Journal Requirements

Response: We have already checked.

2. Please state what role the funders took in the study.

Response: the funders had no role in study design, data collection and analysis, decision to publish, or preparation of the manuscript.

3. Please confirm at this time whether or not your submission contains all raw data required to replicate the results of your study.

Response: Yes, our submission contains all raw data required to replicate the results of our study.

4. Please review your reference list to ensure that it is complete and correct.

Response: We have already reviewed the references and added the doi.

• Reviewer 1

1. The abstract is too long and needs to be revise. The authors should delete the general introduction in abstract. In particular, lines 20 to 25 ‘Francisella tularensis is the causative agent of the zoonotic disease tularemia. F. tularensis is a gram-negative, facultative intracellular bacteria. F. tularensis subsp. (F. novicida) is a facultative intracellular pathogen that proliferates within macrophages; it has the minimum incidence of virulence in humans but is pathogenic in mice. Furthermore, F. novicida shares considerable similarities with highly virulent subspecies and is extensively utilized as a surrogate in the investigation of Francisella.’ should be deleted or moved to the Introduction section.

Response: We have already revised and deleted it.

2. The format of whole manuscript must be checked. Especially, the titles of figures and references (Italic font, etc.) must be checked before submission.

Response: As you suggested, we have already checked.

3. The authors found that F. novicida FTN_0096 mutant could not escape from the phagosome and FTN_0096 is an essential factor for the intracellular replication of F. novicida within host cells. A more detail description of this finding is needed in the Discussion section to help readers better understand the significance of the study. Response: More detailed description has already been added at lines 481-495.

• Reviewer 2

Figs.: "FTN_0096" is probably used as an abbreviation for the FTN_0096 mutant (E12-3), but this is not described anywhere. Please revise this point.

Response: We have already revised it in each figure legend (fig.2-7).

Line 1: The involvement of FTN_0096 in F. novicida pathogenicity has already been reported. Moreover, it is unclear whether FTN_0096 itself is cytotoxic. Therefore, the reviewer suggests considering a revision of the title.

Response: We have already revised the title.

Lines 28, 77: The reviewer does not understand why the authors mentioned the difference in genetic manipulability between HeLa cells and macrophages. Since no gene editing was done in this study, this explanation may be unnecessary.

Response: We have already deleted it.

Lines 35-37, 260-266, 491: Why do the authors consider that the FTN_0096 mutant is captured by autophagosomes at the late stage of infection if this mutant fails to escape from phagolysosomes?

Response: As you suggested we do not have any evidence related to autophagy. Therefore, we have already revised all sentences related to autophagy(lines 32-33, 255-277, 362-381, 480-495)

Lines 48-87, 194, 252, 429-555: In this context, what does “Francisella” refer to? It may be used to indicate all F. tularensis subspecies, rather than the genus Francisella. Please clarify the definition.

Response: We have already changed it at lines 24, 44-82, 197, 441-537.

Lines 100, 196-199: Why were HeLa–FcγRII cells used for screening? In the Introduction, these cells are merely described as a new model. Please explain the advantages of this model and clarify why it was selected for the screening.

Response: We have already added the explanation at lines 95-97, 198-199. At first, we thought we can identify different pathogenic factors form the model using macrophage cell lines, but the results were quite similar to the macrophage model. However this results indicated that this HeLa model mimics macrophage cells very well, and HeLa model is useful to investigate host factors related to Francisella infection in further study. To explain this we made some sentences in discussion (line 443-451)

Line 118: Please describe the screening procedure.

Response: We have already described it at lines 125-128.

Line 154: The use of "strains" may not be appropriate. Please correct.

Response: We have already deleted it.

Line 159: Please provide the full name of PMA.

Response: We have provided it at line 160.

Line 161: Please revise the sentence to indicate that mouse serum contains anti-F. novicida antibodies.

Response: We have already revised it at line 163.

Lines 204, 440: Please revise "FcR HeLa cells" to " HeLa–FcγRII cells".

Response: We have already revised it at lines 209, 448.

Lines 211–213: The reviewer infers that the cells were infected individually with each mutant, and that panel B is a representative image of the screening assay. If this is correct, please revise the sentence.

Response: The image of panel B was from cells infected E12-3 transposon mutant. Therefore we have already revised it at lines 216-218.

Lines 222–223: Using "mutation" instead of "mutant" in this title might be more appropriate.

Response: We have already changed it at line 227.

Line 224: Maybe, a modification like "To analyze the function of FTN_0096" is appropriate.

Response: We have already changed it at line 229.

Lines 252–253: The meaning of this sentence is unclear. You might revise the phrase "particularly the mechanism of FTN_0096 escapes" to "particularly in the escape" for clarity.

Response: We have already revised the whole paragraph, as another reviewer suggested.

Lines 335-345: The authors describe the cytotoxicity of F. novicida against THP-1 cells; however, the results only demonstrate intracellular growth. Is F. novicida cytotoxic to THP-1 cells, and is FTN_0096 involved in this process?

Response: As you suggested, we only observed cytotoxicity by microscopic observation, because THP-1 showed high LDH release after PMA treatment. Therefore, we have already deleted the word “cytotoxicity” at lines 340-349.

Lines 358-359: This sentence is grammatically incomplete. Please revise it.

Response: We have already revised the whole paragraph, as another reviewer suggested.

Which results are the values "50%-75%" and "59%-84%" derived from?

Response: We made the mistake and we have already revised it at lines 481-495.

Lines 491-494: The reviewer could not follow the intention of these sentences. What do the phrases "These findings," "The same results," and "These results" refer to?

Response: We have already changed it (line 509-510).

Lines 501-502: The reviewer could not understand the necessity of this sentence. Could you consider removing it?

Response: We have already removed it after citation[37].

Lines 518, 545: Please clarify what "These results" refer to.

Response: These results refer to the result of bacteria that colocalization in mitochondria and golgi complex. We have already revised them at line 535-537, 570-571, and moved original sentences after the result at lines 514-516, 544-547.

Lines 528-529: Are there any contributions of cytophagy or cytotoxicity to the transport function of the Golgi complex? If so, please provide a citation.

Response: We have already added sentences at lines 551-554 and for reference is no. 46.

Lines 549, 550: This sentence might not match the purpose and results of this manuscript. You might revise this sentence like "In conclusion, we screened a transposon mutant library using HeLa–FcγRII cells and identified FTN_0096 as a pathogenic factor of F. novicida ".

Response: We have already revised it at lines 579-580.

Lines 550-552: Are the authors suggesting that FTN_0096 mainly facilitates the escape of F. novicida from the capture by mitochondria and the Golgi complex? Is there any hypothesis about how FTN_0096 contributes to this escape, and how it is involved in the intracellular growth and cytotoxicity of this bacterium? A discussion of these points might help clarify the manuscript.

Response: We do not suggest FTN_0096 directly facilitates escape from the mitochondria or Golgi complex. Our data imply that FTN_0096 may indirectly affect organelle-associated host responses. We have already added the discussion at lines 572-578.

Reviewer 3

Specific Comments

Methods: The method for calculating the colocalization rate of bacteria and each cell organelle is not provided. Please revise this point.

Response: We have already revised at lines 185-187.

Line 159: Please provide the full name of PMA.

Response: We have provided it at line 160.

Line 161: Please revise "with mouse serum" to "with mouse serum contains anti-F. novicida antibodies".

Response: We have already revised it at line 163.

Lines 204, 440: Please replace "FcR HeLa cells" with "HeLa–FcγRII cells".

Response: We have already revised it at lines 209, 448.

Lines 200-202, 211–213: In Fig. 1B, the bacterial strain that infected the HeLa-FcγRII cells is not clearly described. The reviewer assumes that HeLa-FcγRII cells were infected with the FTN_0096 mutant strain; is this correct? Please revise the legend of Fig. 1B to include more detailed information about the bacterial strain employed.

Response: The image of panel B was from cells infected E12-3 transposon mutant. Therefor we have already revised it at lines 216-218.

Lines 211, 213: "Scale bar = 20 μm" is missing a period, please correct.

Response: We have already revised it at lines 216, 218.

Lines 335-345, 347-348: The authors describe the 'cytotoxicity of the FTN_0096 mutant in THP-1 cells'; however, the cytotoxicity assay was only performed using HeLa–FcγRII cells. Furthermore, results of the cytotoxicity assay for THP-1 cells are neither presented nor discussed in the manuscript. Therefore, the term 'cytotoxicity' should be removed from these sentences.

Response: As you suggested, we only observed cytotoxicity by microscopic observation, because THP-1 showed high LDH release after PMA treatment. Therefore, we have already deleted the word “cytotoxicity” at lines 340-349.

Lines 491-494: The reviewer was unable to clearly understand the intent of these sentences. It is unclear what is meant by the phrases 'These findings,' 'The same results,' and 'These results.'

Response: We have already changed it in lines 509-510.

Lines 549-550: The reviewer finds this sentence confusing. In the context of this study, it would be more appropriate to revise 'and identified FTN_0096 as a pathogenic factor of F. novicida' instead of 'to identify FTN_0096 as a pathogenic factor of F. novicida'.

Response: We have already revised it as you suggested.

---

## [Decision Letter · Decision Letter 1]

Identification of the Francisella novicida FTN_0096 as a factor involved in intracellular replication and host response

PONE-D-25-23563R1

Dear Dr. Shimizu,

We’re pleased to inform you that your manuscript has been judged scientifically suitable for publication and will be formally accepted for publication once it meets all outstanding technical requirements.

Kind regards,

Yung-Fu Chang

Academic Editor

PLOS ONE

Additional Editor Comments (optional):

Reviewers' comments:

Reviewer's Responses to Questions

**Comments to the Author**

Reviewer #1: All comments have been addressed

Reviewer #2: All comments have been addressed

Reviewer #3: All comments have been addressed

2. Is the manuscript technically sound, and do the data support the conclusions?

Reviewer #1: Yes

Reviewer #2: Yes

Reviewer #3: Yes

3. Has the statistical analysis been performed appropriately and rigorously?

Reviewer #1: Yes

Reviewer #2: Yes

Reviewer #3: Yes

4. Have the authors made all data underlying the findings in their manuscript fully available?

Reviewer #1: Yes

Reviewer #2: Yes

Reviewer #3: Yes

5. Is the manuscript presented in an intelligible fashion and written in standard English?

Reviewer #1: Yes

Reviewer #2: Yes

Reviewer #3: Yes

Reviewer #1: Authors identified the virulence factor of Francisella novicida and characterize its roles in phagocytes in this study. All comments were addressed and revised in the whole manuscript point by point. It may be accepted in this journal.

Reviewer #2: (No Response)

Reviewer #3: (No Response)

**Do you want your identity to be public for this peer review?** For information about this choice, including consent withdrawal, please see our Privacy Policy

Reviewer #1: **Yes: ** Suk Kim

Reviewer #2: No

Reviewer #3: No
